# Effect of Polydopamine Coating of Cellulose Nanocrystals on Performance of PCL/PLA Bio-Nanocomposites

**DOI:** 10.3390/ma16031087

**Published:** 2023-01-27

**Authors:** Ivan Kelnar, Ludmila Kaprálková, Sabina Krejčíková, Jiří Dybal, Michaela Vyroubalová, A. M. Abdel-Mohsen

**Affiliations:** Institute of Macromolecular Chemistry, Czech Academy of Sciences, 16200 Praha, Czech Republic

**Keywords:** poly(lactic acid), poly(ε-caprolactone), cellulose nanocrystals, polydopamine coating

## Abstract

In bio-nanocomposites with a poly(lactic acid) (PLA)/poly(ε-caprolactone) (PCL) matrix with neat and polydopamine (PDA)-coated cellulose nanocrystals (CNCd), the use of different mixing protocols with masterbatches prepared by solution casting led to marked variation of localization, as well as reinforcing and structure-directing effects, of cellulose nanocrystals (CNC). The most balanced mechanical properties were found with an 80/20 PLA/PCL ratio, and complex PCL/CNC structures were formed. In the nanocomposites with a bicontinuous structure (60/40 and 40/60 PLA/PCL ratios), pre-blending the CNC and CNCd/PLA caused a marked increase in the continuity of mechanically stronger PLA and an improvement in related parameters of the system. On the other hand, improved continuity of the PCL phase when using a PCL masterbatch may lead to the reduction in or elimination of reinforcing effects. The PDA coating of CNC significantly changed its behavior. In particular, a higher affinity to PCL and ordering of PLA led to dissimilar structures and interface transformations, while also having antagonistic effects on mechanical properties. The negligible differences in bulk crystallinity indicate that alteration of mechanical properties may have originated from differences in crystallinity at the interface, also influenced by presence of CNC in this area. The complex effect of CNC on bio-nanocomposites, including the potential of PDA coating to increase thermal stability, is worthy of further study.

## 1. Introduction

It is now well accepted that the impact of nanofillers (NFs) on nanocomposite performance is greater in the case of a multicomponent polymer matrix [1,2,3,4]. The reason is a more complex effect of NFs that combines reinforcement with effective compatibilization, structural transformations including changes in the interface, and formation of effective hierarchical, e.g., rigid/soft structures [5]. In the case of blends of semicrystalline polymers, the effect of crystallinity can also be considered [6]. Such a combination of blending and nanocomposite concepts is also applied to modify the parameters of biodegradable/biocompatible polyesters [7]. Biodegradable materials with a broad range of mechanical properties are based on a rigid and brittle poly(lactic acid (PLA)/low modulus elastic poly(ε-caprolactone (PCL) blend [8,9,10]. Here, the important issue is to eliminate their low compatibility [8], which can also be achieved by targeted application of various NFs. In addition to modification with various nanosilicates [11,12,13], fair effectiveness was found with graphite nanoplatelets (GNPs) [14] and multiwall carbon nanotubes [15], and interesting results were obtained via in situ linking to graphene oxide [16]. The important role of NFs was demonstrated in the case of microfibrillar composites with PLA fibrils (fibers) formed in situ in the PCL matrix. In addition to dual reinforcement, the effect of NF, such as montmorillonite, halloysite nanotubes, and GNP, is even more complex compared to blends due to the affecting of drawing and special effects of NF migration in the course of drawing/solidification) [17,18]. To avoid compromising the biodegradability/biocompatibility of the nanocomposites [7], application of “organic” NFs, e.g., cellulose nanocrystals (CNC) [19], chitin nanowhiskers (CNW) [20], and silk fibroin [21], has been reported. This also includes the application of CNC and CNW grafted with polymer (mostly PCL or PLA) [22,23,24]. Motlong et al. observed an increase in modulus and toughness in the CNC-modified PLA/PCL 70/30 mixture [19], while important effects of PCL- and PLA-grafted CNC on crystallinity and structure were reported for PLA/PCL 70/30 [22] and PLA/PCL 50/50 [23] systems.

We have shown that the complex effects of GNP are strongly dependent on the polymer blend ratio [14]. These effects of NFs are also influenced by localization controlled by kinetic and thermodynamic effects such as mixing protocols and affinity of NFs to respective components determined by the nature of NFs together with suitable modifications [25]. Especially in a blend-based matrix with a similar affinity between NFs and polymer components, the NF is preferentially localized in the first melting phase, i.e., PCL in the case of PLA/PCL [14]. Moreover, modification of the surface of NFs [26] is a crucial tool to control their dispersion and, thus, reinforcement. Recently, it has been shown that an effective facile “bio”-modification of nanofillers is the polydopamine (PDA) coating, representing a transformation to the biocompatible NF type with quite different surface energy [27,28]. The practically unexplored area in this study is the reveal of the relationship between well-known unique features (e.g., glue effects) of PDA and their potential for both interactions and reactions [29], undoubtedly affecting the structure in multicomponent matrix-based nanocomposites, nucleation, etc. Of importance is also the stabilizing effect of PDA on polymers [30]. To the best of our knowledge, the effect of targeted pre-blending of CNC and CNCd, with a varied ratio of PLA and PCL, which also influences the aforementioned complex NF effects of blend performance, has not yet been studied. Therefore, this study investigates the properties of PCL/PLA/CNC and PDA-coated CNC.

## 2. Materials and Methods

### 2.1. Materials

Poly(lactic acid) (PLA) Ingeo 2002D (Nature Works, Minnetonka, MN, USA) was used with a D-isomer content of 4.3%, Mw of 2.53 × 10^5^ g·mol^−1^, melt flow rate of 6 g/10 min (190 °C/2.16 kg), and density of 1.24 g·cm^−3^. Poly(ε-caprolactone (PCL) CAPA 6800 (Perstorp, Malmö, Sweden) was used with a Mw of 8 × 10^4^ g·mol^−1^, melt flow rate of 3 g/10 min (160 °C/2.16 kg), and density 1.145 g·cm^−3^. Dimethyl formamide (DMF) and dopamine were purchased from Sigma Aldrich (Prague, Czech Republic). BGB Ultra^TM^ cellulose nanocrystals (CNCs) in an 8.0% *w*/*w* aqueous suspension (BGB Inc., Saskatoon, Canada) were used.

### 2.2. Nanocomposite Preparation

For polydopamine coating [27], 400 mg of CNC was dispersed in 400 mL of water with pH ~11 adjusted by Tris buffer. After the addition of 200 mg of dopamine, polymerization proceeded for 24 h at ambient temperature with subsequent washing with water and lyophilization.

To prepare 10% masterbatches, CNC and CNCd isolated by lyophilization were ultrasonically dispersed in DMF. After the addition of PLA or PCL (polymer/CNC ratio of 9/1 *w*/*w*) and mixing at 85–90 °C for 2.5 h, the solvent was evaporated at 50 °C for 24 h.

The nanocomposites were prepared by melt-mixing in a Haake Mini CTW (Thermo Scientific^TM^, Karlsruhe, Germany) at 140 °C (for the blend and PCL matrix nanocomposite) and 190 °C (for PLA) for 5 min at 45 rpm. CNC and CNCd were added as a 10% masterbatch in PCL or PLA. Two sets of samples differing in the addition of CNC via PCL or PLA masterbatch with 80/20, 60/40, 40/60, and 20/80 PLA/PCL weight ratios were prepared. The composition of samples is shown in Appendix A. The masterbatch and granules of both polymers were combined to achieve 2% CNC or CNCd content in the nanocomposites. In the CNC-free samples, PLA or PCL analogously treated with dimethyl formamide were applied. Subsequently, a film of ~0.2 mm thickness was prepared in a laboratory press (140 °C for 5 min).

### 2.3. DFT Modeling of Stabilization Energy between Constituents

For closer insight into the interactions of CNC and polydopamine-modified CNC and polymer components, we carried out model quantum chemical calculations of interactions among the structural units of CNC, CNCd, and the respective polymer chains. The applied structural units are introduced in the Appendix A. Hydrogen-bonding effects were studied at the DFT level of theory with the B3LYP exchange correlation functional in combination with the semi-empirical dispersion correction GD3BJ [31]. Calculations with the 6-31 + G(d,p) basis set were performed using the Gaussian 16 program package (Gaussian 16, Gaussian Inc., Wallingford, CT, USA) [32]. The fully optimized geometries represent the true energy minima on the potential energy surface. Here, no imaginary frequencies were obtained with normal mode calculations. Moreover, in the calculations of the stabilization energies of the hydrogen-bonded complexes (the difference between the energy of two interacting structural units and the complex), the Boys and Bernardi counterpoise correction was applied in order to consider the basis set superposition error [33].

### 2.4. Characterization of Blends Structure

The structure was examined using scanning electron microscopy (SEM) with a Maia microscope (FEI, Brno, Czech Republic). The injection-molded specimens broken under liquid nitrogen were etched in 20% NaOH for 20 min to remove the PLA component in the PCL matrix and bicontinous samples; the PLA matrix samples were etched with THF vapor at 45 °C for 4 min to “visualize” PCL inclusions [34]. The surface of unetched samples is shown in Appendix A. The size of the dispersed particles was investigated using a MINI MOP image analyzer (Kontron Co., Munich, Germany) with manual marking of the circuit of the evaluated particles. At least 200 particles were evaluated in each sample. For the Tecnai G2 Spirit transmission electron microscope (TEM) (FEI, Brno Czech Republic) observations, ultrathin (60 nm) sections were prepared under liquid nitrogen using an Ultracut UCT ultramicrotome (Leica Mikrosysteme GmbH, Wien, Austria).

### 2.5. Testing

Tensile tests were carried out using an Instron 5800 (Instron, High Wycombe, UK) apparatus on dog-bone samples cut from the films at 22 °C with a crosshead speed of 1 mm/min (ISO 527-2). At least eight specimens were tested for each sample. Young’s modulus (*E*), stress at break (*σ*), and elongation at break (*ε*_b_) were evaluated; the corresponding variation coefficients did not exceed 10%, 2%, and 20%, respectively.

Tensile impact strength, *a*_t_, was measured using an unnotched injection-molded and CEAST Resil impact junior hammer (CEAST S.p.A., Torino, Italy) with an energy of 4 J (variation coefficient 10–15%). The reported values are the averages of 10 individual measurements.

The DSC analysis was performed using a TA Instruments Q2000 DSC apparatus (TA Instruments, New Castle, DE, USA). The measurements were carried out in a heating–cooling–heating regime between 0 °C and 200 °C at a constant heating and cooling rate of 10 °C/min. The values of 139.5 J/g and 93.7 J/g were used as the melting enthalpies of 100% crystalline PCL and PLA, respectively. The final crystallinity values were calculated in relation to the real weight fraction in the sample.

The rheological characterization was carried out using an ARES apparatus (Rheometric Scientific, Piscataway, NJ, USA) with the parallel-plate geometry at 170 °C using an oscillatory shear deformation within the frequency range of 0.1–100 rad/s. The amplitude of oscillation was 1%.

Dynamic mechanical analysis (DMA) was performed in single-cantilever mode using a DMA DX04 T apparatus (RMI, Pardubice, Czech Republic) at 1 Hz and heating rate of 1 °C/min from −120 to 150 °C.

## 3. Results and Discussion

### 3.1. CNC and Polydopamine-Coated CNC Localization

Due to the low reliability of surface tension evaluation and, thus, wetting coefficient assessment, we applied an alternative approach. As also indicated by our recent results [35], DFT may provide a good assessment of affinity between components by evaluating the binding energy. Appendix A show a similar binding energy between PLA/CNC and PCL/CNC, which indicates low tendency of CNC for preferential localization. This is in contrast to polydopamine-coated CNC (CNCd) where markedly higher affinity to PCL was found compared to that for PLA. This also corresponds to fair dispersion of CNCd in PCL, whereas, in PLA, ordering to fibrous arrays occurs (Figure 1a). At the same time, the ordering of CNCd is also supported by higher mutual interaction energy between CNCd compared to CNC (Appendix A). This performance of CNCd is further documented by differences between the TEM and SEM images of neat CNC and CNCd (Appendix A). All these facts confirm the different characteristics of PDA-coated CNC. Because of the similar affinity of CNC to both polymers, CNC localization is obviously determined by initial pre-mixing in respective components, whereas a marked effect of migration on the PCL phase may be expected with polydopamine-coated CNC. This is undoubtedly also influenced by kinetic factors [36] especially by the relatively lower viscosity of PLA due to DMF treatment (Appendix A). Especially in the case of low content of the polymer component with preblended CNC (80/20 and 20/80 ratio), we can also expect some extent of NF migration and presence at the interface. Unfortunately, single nonordered CNCs are invisible in TEM [23].

In spite of the low resolution of Figure 1b,c (which was partly eliminated by evaluation of tens of images), showing TEM images of the 80/20 system with CNCd, we might consider a marked difference between both structures. In the case of CNCd pre-blended in PLA, ordered fibrous arrays CNCd are present inside the PLA matrix (Figure 1b), similarly to analogous PLA/CNCd nanocomposite (Figure 1a). This is probably also a consequence of expected hindered migration of ordered “nanofibers” (negative effect of larger size and aspect ratio [14]), while, in the case of preblending in PCL, most CNCd probably remains in dispersed PCL (Figure 1c). The fact that CNC localization is predominantly determined by pre-dispersing in respective phases also follows from structural alterations in the bicontinuous 60/40 and 40/60 PLA/PCL systems (see below) prepared using the PCL/CNC or PLA/CNC masterbatches. In addition to different effects of components’ viscosity (Appendix A), the changed structures with CNCd may be aided by more marked migration and presence at interface in the case of the PLA/CNCd masterbatch (see Section 3.2).

### 3.2. Structure of Blends and Related Nanocomposites

From Table 1, it follows that the different mixing protocols in neat blends with 80/20 and 20/80 PLA/PCL weight ratio, including the treatment of PLA or PCL with DMF (analogous to masterbatch preparation), also lead to different size of inclusions. This is undoubtedly a consequence of the unexpected effect of DMF treatment on the viscosity of polymer components. The rheological characterization (Appendix A) reveals that, in the case of PLA, a relatively marked drop occurs, whereas, in PCL, this leads to slightly increased viscosity. In the case of the 80/20 system, this is reflected in the larger size of the PCL inclusions in the blend with PCL_dmf_ due to hindered break-up because of higher viscosity (Table 1). The lover size with PLA_dmf_ indicates a less marked negative effect of the lower viscosity of the PLA matrix on the evolution of the structure. In the case of nanocomposites, we must bear in mind that these changes in viscosity undoubtedly also influence the migration of NF between phases [36].

In the case of the nanocomposite with 80/20 PLA/PCL ratio, the addition of CNC leads to a smaller size of PCL inclusions than in related neat blends, which indicates a compatibilizing effect of CNC. This refinement is slightly higher if CNC is added as the PCL/CNC masterbatch (Table 1), indicating that the expected negative effect of the presence of CNC in PCL on particle breakup is apparently compensated for by the higher viscosity [36] of “solvent-processing-free” PLA (Appendix A). Here, the cutting effect of the expected CNC stacks [37] on PCL breakup and presence of CNC at the interface may also be considered.

The lowest particle size with CNCd indicates its higher compatibilizing effect, as also supported by the higher viscosity of PLA/CNCd (Appendix A), with more marked migration to PCL expected (due to higher affinity indicated by DFT) and, thus, a possibly higher presence at the interface, hindering the coalescence. The similar particle size with PCL/CNCd or PLA/CNCd pre-blends (Table 1) further confirms the different “structure-directing” ability of CNCd.

The effect of CNC and CNCd on the size of inclusions in the systems with 20/80 PLA/PCL ratio is shown in Table 1 and Figure 2. Here, the effects of the DMF treatment mentioned above (Appendix A) have an opposite impact on size in the neat blend, i.e., a higher viscosity of the PCL matrix leads to smaller PLA inclusions, which exceeds the favorable effect of DMF-induced lower PLA viscosity on particle breakup. The addition of CNC using both mixing protocols leads to a smaller size with a trend similar to the neat blend. Here, in addition to the compatibilizing effect, we can consider the dominancy of the higher viscosity of the PCL/CNC matrix. With CNCd, the most marked reduction in size also indicates its higher compatibilizing activity. In this case, the slightly smaller size for the PLA masterbatch probably corresponds to positive effect of higher migration of CNCd to PCL and, thus, a more marked presence at the interface and an increase in matrix viscosity.

From Figure 3 and Figure 4, it is obvious that, in systems with a bicontinuous structure, the mentioned effects of the mixing sequence on viscosity of components and NF localization predominantly affect dynamic asymmetry between the components [38]. This is reflected by a change in the degree of continuity of respective phases and different content of the subinclusions inside them, while the inherent compatibilizing effect (supported by interfacial localization) leads to the refinement of continuous “threads”.

In the case of the neat 60/40 PLA/PCL blend (Figure 3a,b), treatment of PLA with DMF (reduced viscosity) decreases continuity of the PLA phase, as well as leads to more PCL subinclusions (Figure 3a). Addition of CNC to PLA (using PLA/CNC masterbatch) leads to higher continuity of PLA with low content of subinclusions (Figure 3c), whereas PCL/CNC leads to different finer morphology with lower continuity of the PLA phase with relatively high content of partially interconnected subinclusions (Figure 3d), probably inside both continuous phases. In other words, this structure shows some type of dual bicontinuity where each bulk thread contains a finer bicontinuous structure (Figure 3d). This is most probably caused by a joint effect of both DMF treatment and CNC-induced changes in the rheological behavior of components (Appendix A) on dynamic asymmetry.

In the case of the PLA/CNCd masterbatch (Figure 3e), we can see a higher content of PCL subinclusions in comparison with CNC. Here, the positive effect of the higher viscosity of PLA is accompanied by migration of CNCd to PCL and the expected presence of CNC at the interface. Moreover, higher affinity between PCL and CNCd may lead to the formation of some complex structures (e.g., core/shell [5] structures) inside the PLA phase. As expected, higher continuity of the PCL phase can be observed with the PCL/CNCd masterbatch (Figure 3f), including a quite different structure of threads with a marked reduction in subinclusions, compared with the analogous CNC-modified system.

Figure 4 shows the structure of the systems with 40/60 PLA/PCL ratio. The lower viscosity of PLA_dmf_ (Figure 4a) leads to loss of continuity with the marked presence of rough PLA inclusions, whereas the application of PCL_dmf_ in combination with the higher viscosity of untreated PLA leads both to higher continuity and to a finer structure of the PLA phase (Figure 4b). Of interest is that application of both CNC-masterbatch types (Figure 4c,d) leads to similar structures with increased continuity of the PLA phase in contrast to the dominating continuity of PCL in CNC-free blends (Figure 4a,b). This also confirms the marked effect of blend ratio [14].

In the case of the PLA/CNCd masterbatch (Figure 4e), the threads of PLA are finer than with CNC, but we can also see also formation of numerous PLA subinclusions in the PCL phase at the expense of PLA threads. This can be considered as a finer “compatibilized” structure of an analogous neat blend, which corresponds to the higher compatibility effect of CNCd, supported by the expected higher presence at the interface. With the PCL/CNCd masterbatch application (Figure 4f), a fine structure, albeit with unexpected higher continuity of the PLA phase in comparison with PLA/CNCd masterbatch, is observed. At the same time, PCL threads also contain a relatively high content of elongated PLA subinclusions. This further confirms differences in the performance of CNC and CNCd. We consider the effect of higher CNCd migration from the PLA phase to be important.

### 3.3. Mechanical Properties

Table 2 shows the comparable reinforcing effect of both CNC and CNCd on single PLA and PCL, which is reflected mainly in the increase of modulus, more marked for soft PCL. At the same time, Figure 5a–d shows the significant difference in mechanical properties of the PLA/PCL systems for all polymer ratios, while also indicating the crucial effect of the mixing protocol on the mechanical properties. In the case of PLA/PCL with a 80/20 ratio, we can see the fair reinforcing effect of CNC and higher *E* with the PLA masterbatch application, i.e., matrix reinforcement. The relatively high modulus corresponds predominantly to a high content of rigid brittle PLA. The reduction in *E* in the system with the PCL/CNC masterbatch undoubtedly consists of a lower contribution of the reinforced minority PCL phase with a lower volume. According to the Kerner model [39]*,* the 10% increase in *E* for PLA matrix leads to its ~200 MPa increase in the mixture, while an analogous increase in *E* in PCL leads to a gain of only ~10 MPa in the blend. This discrepancy cannot, thus, be eliminated by the mentioned higher reinforcing effect of CNC in soft PCL (Table 2). Moreover, this feature confirms the low extent of CNC migration to the PLA phase (see above). For example, the lower *E* in the case of CNCd may be caused by the fact that the PDA coating acts as a soft interface [40], but we may also consider its different effect on crystallinity at the interface in comparison to CNC. The marked drop of *E* with the PCL/CNCd pre-blend indicates the presence of some still unexplained antagonistic effects, as also observed by other authors [41,42], which are worth studying. An important feature of this structure is that it leads to higher toughness.

In the case of tensile strength (TS), the addition of CNC leads to a negligible decrease, similarly to an analogous system with other NF [26], whereas a more marked drop with CNCd probably also corresponds to the mentioned antagonistic effects (Figure 5b). The reduction in elongation of the nanocomposites is in accordance with reinforcement (Figure 5c).

An important result is that toughness (Figure 5d) in 80/20 nanocomposites is increased despite the predominantly harmful effect of NF on this parameter [26], including the hindrance of cavitation of PCL inclusions. High toughness only in the case of PCL/CNC masterbatch (with expected higher CNCd content due to lower migration to PLA) indicates that the possible presence of CNC stacks inside PCL inclusions may lead to favorable intraparticle debonding-induced cavitation [43]. At the same time, analogously to other successful systems [5,14], we consider enhancing the toughening ability of PCL inclusions by forming complex structures such as PCL core/shell particles with CNC at the surface. This can lead to alteration of PLA crystallinity near the interface (e.g., arrangement with lower resistance against shear flow) increasing the extent of PCL cavitation-induced energy-absorbing micro-deformations of the surrounding matrix [44]. The fact that only the systems with lower *E* have higher toughness probably indirectly confirms changes in the interface, having a marked negative effect on stiffness.

In the case of PCL with dispersed PLA, i.e., 20/80 ratio (Figure 2), reinforcement with CNC has a similar trend to NC with the PLA matrix, but with a marked difference of higher *E* also for CNCd, especially with the PCL/CNCd masterbatch (which leads to a decrease in *E* for 80/20 NC). It is clear that the reinforcing effect here undoubtedly exceeds the possible structural changes. The different role of CNCd is further confirmed by the high value of TS, especially for the PLA/CNCd masterbatch, together with the fact that substantially lower elongation is not accompanied by a reduction in toughness, which is comparable to other systems. The slightly decreased toughness of NC is more marked in the PLA/NF pre-blends, i.e., in the presence of more rigid PLA inclusions. This corresponds to the fact that the toughness of a system with ductile matrix/brittle inclusions may be supported by plastic deformation of PLA inclusions [45], which is hindered by NF. This effect apparently exceeds the positive effect of NF, i.e., altering of the ratio of component parameters and enhancing interfacial bonding (facilitating stress transfer) by the presence of NF at the interface [46].

In the case of PLA/PCL with a 60/40 ratio (Figure 3), variations of *E* in the systems prepared corresponds to changes in continuity of both components and different content of subinclusions (Figure 3) caused by different localization of NFs (See Section 3.2), which can eliminate reinforcement. Here, a well-known fact is of importance [38], i.e., that the impact of the continuous rigid strong component (PLA) on stiffness is greater in comparison with its presence as dispersed inclusions. For example, for 40% PLA content, the modulus of co-continuous structure calculated by the Davies model [47] is ~20% higher than that for matrix inclusions (using the Kerner model [39]). The highest *E* in the system with the PLA/CNCpre-blend corresponds to high continuity of PLA without PCL subinclusions (Figure 3c), whereas higher continuity of the weak PCL phase leads to a significant drop in the case of the PCL/CNC masterbatch (Figure 3d). In the case of the PLA/CNCd pre-blend, lower *E* in comparison with the analogous CNC system results from a higher content of the PCL subinclusions inside the PLA threads (Figure 3e), whereas the much more marked drop for PCL/CNCd is connected with markedly increased continuity of the PCL phase with PLA inclusions at the expense of continuous PLA (Figure 3f). If we take into account that a similar decrease was found for the 80/20 system*,* the antagonistic effects mentioned above may also participate in this behavior. Higher values of tensile strength for neat and CNC-modified blends with higher continuity of the PLA phase indicate an important effect of this structural transformation on this property, while much lower values for both CNCd-modified systems correspond to less favorable rough structures with subinclusions. Elongation (Figure 5c) is mostly reduced by CNC but a relatively high value in systems with more continuous PCL (using PCL/CNC pre-blend) indicates that this effect can also be eliminated by transformation of the structure. As shown in Figure 5d. it is obvious that toughness is not reduced by NF. The practically zero correspondence to the trends of other properties indicates that the energy absorption is determined by numerous, probably also controversial, effects in the co-continuous systems with subinclusions and different dimensions of threads (Figure 3 and Figure 4). The relatively high value for the system with the PLA/CNCd masterbatch indicates a positive effect of the PCL inclusions in PLA thread (Figure 3e).

With the 40/60 PLA/PCL ratio (Figure 4), we found the best *E* for the PLA/CNC masterbatch and a mostly similar difference between the respective systems to that for 60/40 ratio (Figure 5a). Thus, in this system with a bicontinuous structure, the alterations of morphology control the contribution of components. At the same time, at this blend ratio, the structural changes induced by the mixing protocol are relatively low, especially for CNC. Therefore, the differences in *E* between respective systems are lower in comparison with the 60/40 system. The most marked difference from the 60/40 system is the higher *E* for PCL/CNCd (in contrast to PLA/CNCd), which corresponds to the unexpected higher continuity of the PLA phase (Figure 4f).

The strength of NC is slightly lower than that of the neat blends. The markedly higher values for DMF-treated PCL and PCL/CNC correspond to finer dimensions of threads (Figure 4b).

Elongation is also reduced by NF; the relatively marked difference also confirms the important role of structural transformations. Of interest is the highest value of toughness for PCL/CNCd, despite the fact that the continuous PCL phase contains a relatively high amount of the PLA inclusions. Here, we can consider a possible contribution of their plastic deformation in this ductile/brittle phase.

### 3.4. Crystallinity of Blends and Nanocomposites

In this work, we focused on the crystallinity of the samples used for mechanical testing; that is, the DSC evaluation of “virgin” as-prepared samples was applied.

The crystallinity of PCL and PLA in dependence on the blend ratio in the samples prepared using the PLA masterbatch is shown in Figure 6a, while Figure 6b shows results for the analogous PCL masterbatch-prepared samples. We can see that, with increasing PLA content, the structural changes from the PCL matrix/PLA inclusions (Figure 2) to the bicontinuous structure (Figure 3 and Figure 4) and the PLAmatrix/PCL inclusions have a relatively low impact on the crystallinities of both components. The moderate increase in PCL crystallinity with increasing PLA content, i.e., the positive effect of blending [14] apparently corresponds to an increase in the area of interface by transformation from (bi)continuous PCL threads to inclusions. A similar, but less marked effect is found for an analogous structural transformation of PLA with increasing PCL content. In the case of PCL inclusions (80/20 PLA/PCL ratio), we can see their higher crystallinity for both CNC- and CNCd-modified systems, which indicates their comparable nucleating effect. At the same time, the lower crystallinity of the neat blend is still higher than that of the bulk PCL. This performance seems to consist of the elimination of the possible negative effect of confinement [48] due to small dimensions ~250–600 nm of the inclusions (Table 1) by other effects, including the mentioned blending with PLA [14]. Lastly, the nucleation effect of CNC and CNCd is relatively low, which is indicated by the presence of a single melting peak of PCL in contrast to CNC grafted with polyesters [22].

From Figure 6a,b, it further follows that CNC and CNCd reduce the initial crystallinity of PLA while causing a higher extent of cold crystallization (CC). This indicates some hindering effect of NFs, causing a deceleration of chain mobility upon cooling in the press, which is accompanied by some nucleation effect of CNC and CNCd on CC. This effect is more marked in the case of application of both NFs as a masterbatch in PLA, i.e., with expected higher NF content in PLA. The slightly lower effect of CNCd probably corresponds to its higher migration to the PCL phase (see above).

Generally, the differences in crystallinity among systems with CNC, CNCd, and the neat blend, including the mixing protocol, are low but with relatively marked dependence on the blend ratio, i.e., for PCL. The presence of NF at the interface may lead to practically indetectable variations in crystallinity in this area with a possible significantly high impact on the mechanical properties.

### 3.5. Dynamic Mechanical Analysis

In the case of single components (Table 2), the addition of CNC leads to a slight decrease in *T*_g_ (~0.5 °C) for PLA and a more marked decrease for CNCd. A similar effect was found for organophilized montmorillonite and halloysite nanotubes [5]. At the same time, with PCL, a slight increase in *T*_g_ and a more marked increase with CNCd was found. This corresponds to the still not well explained effect of NF on this behavior [49] and confirms the different performance of CNCd. Figure 7a,b shows the temperature dependence of the loss modulus in the systems with a bicontinuous structure, i.e., 60/40 and 40/60 PLA/PCL ratios, with relatively marked NF-induced changes in structure (Figure 3 and Figure 4). The low increase in the low-temperature peak corresponding to the *T_g_* of PCL and the parallel low shift of PLA peak at ~60 °C to lower temperatures indicate the limited “compatibility” effect of CNC and CNCd. At the same time, in the case of a relatively complex bicontinous structure (Section 3.2), cumulation of more effects affecting *T_g_* is anticipated, including micromechanical transitions [50] that originate in multiphase structure and changes in the interface parameters. A certain decrease in *T*_g_ of PLA in all systems is a consequence of PCL dissolved in the PLA-rich phase [51]. This effect is coupled with the mentioned negative effect of CNC and CNCd on *T_g_* of PLA in the case of the PLA masterbatch application, which is in accordance with the relatively higher *T_g_* of PLA in the case of PCL/NF masterbatch and, thus, much lower NF content in the PLA phase. At the same time, a dissimilar trend in the 40/60 composition (lower *T_g_* of PLA with PCL/CNC against PLA/CNC masterbatch) most probably corresponds to differences in CNC- and CNCd-induced structures (Figure 3 and Figure 4), including the interface and related micromechanical transitions. This peculiar performance is also indicated by the thermal dependence of the storage modulus shown in Appendix A (e.g., even crossing of curves); however, a detailed explanation of these effects is outside the scope of this work.

## 4. Conclusions

From the results obtained, it follows that the performance of the PLA/PCL/CNC nanocomposites with 80/20, 60/40, 40/60, and 20/80 PLA/PCL ratios is significantly influenced, in addition to reinforcement, by the CNC-induced changes in structure. This structure-directing ability is significantly influenced by the polydopamine coating of CNC, which also increases the affinity to PCL, whereas interactions of neat CNC with both PLA and PCL are similar. Moreover, we observed a crucial effect of the CNC addition protocol using solution-prepared pre-blends with PLA or PCL on the performance of the nanocomposites. In the matrix/inclusion systems with 80/20 and 20/80 composition, the mostly improved mechanical performance originated from favorable nanofiller-induced changes in the structure, interface, and properties of components. This could lead to an increased toughening ability of the PCL inclusions or an energy-absorbing plastic deformation of the PLA inclusions, whereas some antagonistic effects were found with CNCd application.

In the systems with a dominating bicontinuous structure (60/40 and 40/60 ratio), the structure and mechanical properties are more markedly influenced by the mixing protocol and, thus, by NF localization in respective phases. This leads to an effect on the dynamic asymmetry between components and related changes in their continuity. Due to the different properties of both components, strength and stiffness, in particular, are markedly higher with the NF-increased continuity of rigid PLA, whereas an analogous increase in continuity of PCL at the expense of PLA can eliminate the reinforcing effect of NF. The observed differences between CNC and CNCd nanocomposites consist of a different effect on viscosity, along with a compatibility effect controlled by the interfacial localization and degree of migration between phases. With respect to the presence of some subinclusions in the respective continuous phases, further NF-induced effects are also important. Due to the very low differences in the “bulk” crystallinity of the components in the systems studied, the alteration of mechanical properties may be caused by possible (practically undetectable) differences in crystallinity at the interface, which are also influenced by the presence of NF in this area.

## Figures and Tables

**Figure 1 materials-16-01087-f001:**
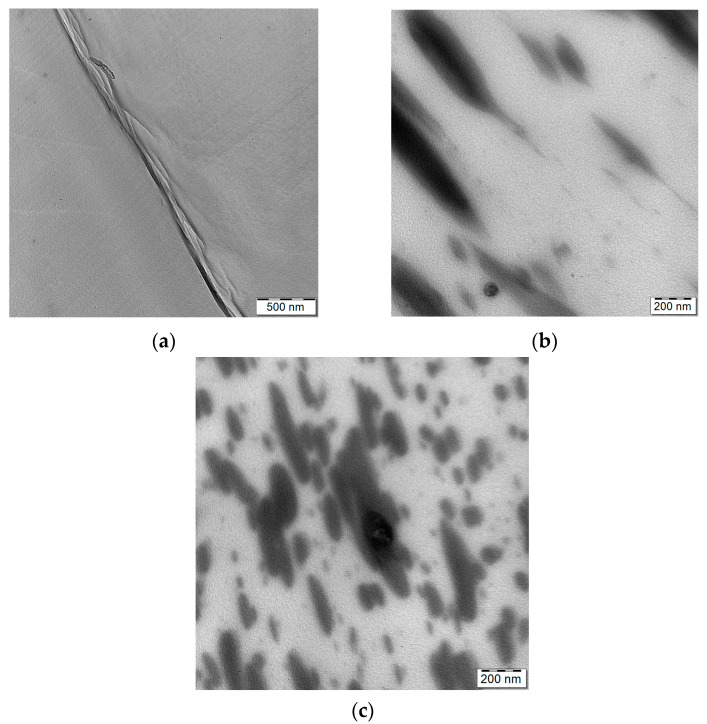
TEM images of fibrous ordered structure of CNCd (**a**) in PLA, and (**b**) in 80/20 PLA/PCL (with PLA/CNCd pre-blend); (**c**) TEM image of 80/20 PLA/PCL (prepared using PCL/CNCd masterbatch).

**Figure 2 materials-16-01087-f002:**
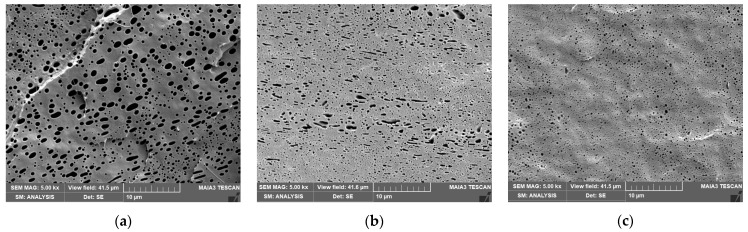
SEM images of PLA/PCL 20/80 with (**a**) PLA treated with DMF, (**b**) PCL/CNC masterbatch, and (**c**) PCL/CNCd masterbatch; the dark voids represent NaOH-etched PLA inclusions.

**Figure 3 materials-16-01087-f003:**
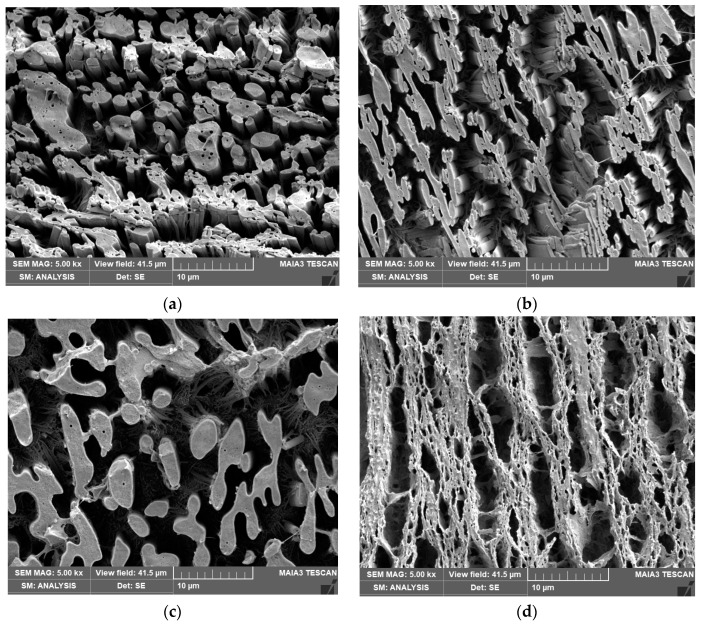
SEM images of 60/40 systems: (**a**) neat blend with DMF treated PLA; (**b**) DMF-treated PCL; NC with (**c**) PLA/CNC masterbatch, (**d**) PCL/CNC masterbatch, (**e**) PLA/CNCd masterbatch, and (**f**) PCL/CNCd masterbatch; the dark area represents the NaOH-removed PLA phase. A structure of the unetched sample (**c**) is shown in Appendix A.

**Figure 4 materials-16-01087-f004:**
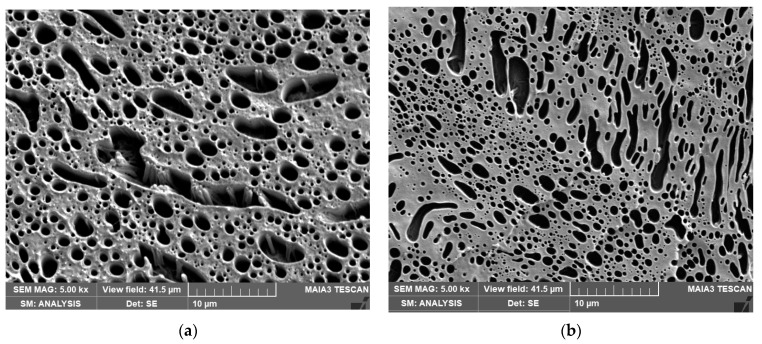
SEM images of 40/60 systems: (**a**) neat blend with DMF treated PLA; (**b**) DMF-treated PCL; NC with (**c**) PLA/CNC masterbatch, (**d**) PCL/CNC masterbatch, (**e**) PLA/CNCd masterbatch, and (**f**) PCL/CNCd masterbatch; the dark area represents the NaOH-removed PLA phase. A structure of the unetched sample (**d**) is shown in Appendix A.

**Figure 5 materials-16-01087-f005:**
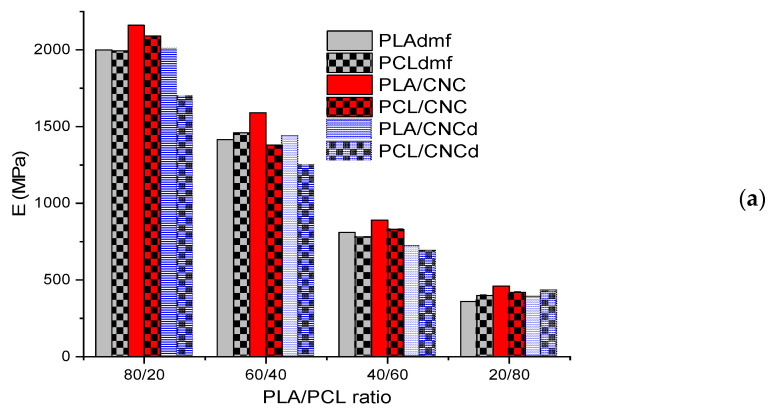
Mechanical properties: (**a**) Young’s modulus; (**b**) tensile strength; (**c**) elongation; (**d**) tensile impact strength.

**Figure 6 materials-16-01087-f006:**
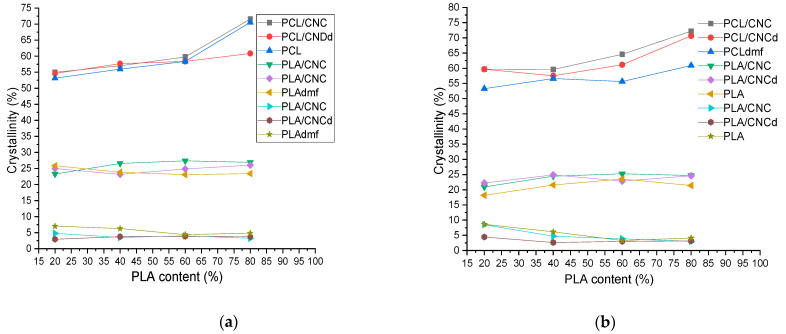
(**a**) Crystallinity of samples prepared using CNC and CNCd masterbatch in PLA, and DMF-treated PLA (PLAdmf); (**b**) Crystallinity of samples prepared using CNC and CNCd masterbatch in PCL, and DMF-treated PCL (PCLdmf);The three sets of plots (from top to bottom) are the crystallinity of PCL, cold crystallization of PLA after heating, and initial crystallinity of PLA, respectively. Each set contains PLA/CNC masterbatch, PLA/CNCd masterbatch, and NF-free blend (PLAdmf).

**Figure 7 materials-16-01087-f007:**
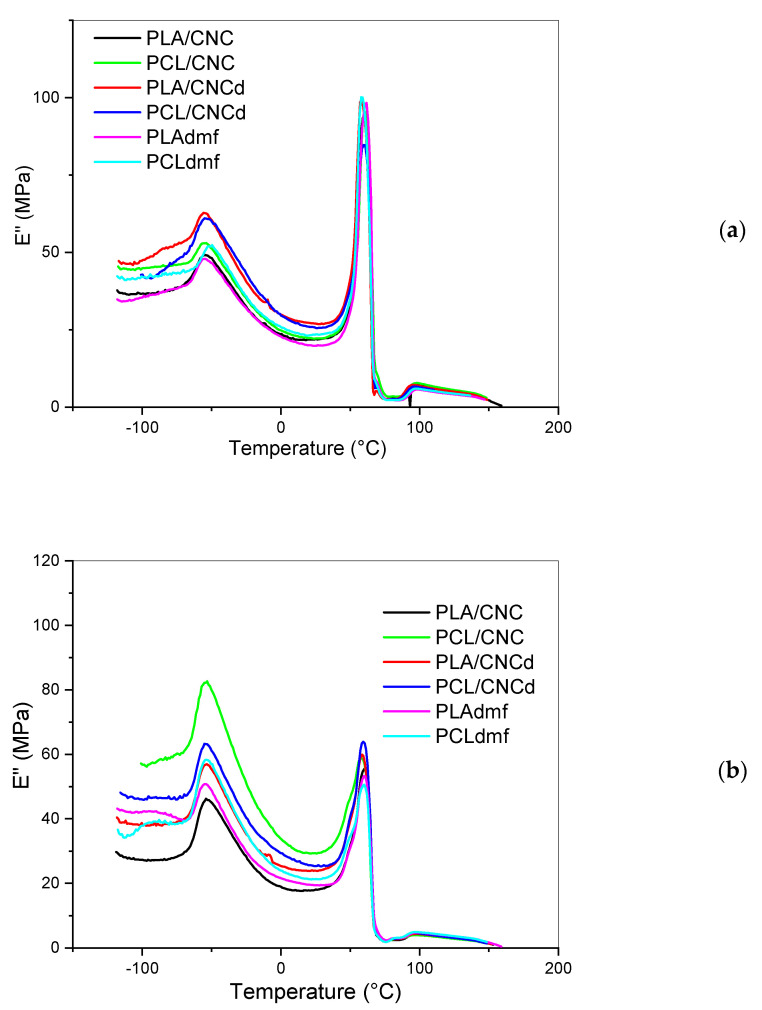
Temperature dependence of loss modulus of (**a**) 60/40 PCL/PLA ratio, and (**b**) 40/60 ratio.

**Table 1 materials-16-01087-t001:** Size of dispersed particles in PLA/PCL blends and nanocomposites with matrix/inclusions structure (PLA_dmf_ and PCL_dmf_ are components treated with dimethyl formamide analogously to masterbatch preparation).

Blend Ratio	PLA/PCL		PLA/PCL/CNC		PLA/PCL/CNCd	
	PLA_dmf_	PCL_dmf_	PLA/CNC	PCL/CNC	PLA/CNCd	PCL/CNCd
80/20 PCL Size (µm)	0.51	0.57	0.39	0.34	0.27	0.27
20/80 PLA Size (µm)	0.6	0.55	0.38	0.33	0.24	0.25

**Table 2 materials-16-01087-t002:** Effect of CNC and CNCd on properties of components.

Composition	E (MPa)	Stress at Break (MPa)	Break Strain (%)	Toughness (kJ/m^2^)	Tg (°C)
PLAdmf	2629 ± 261	48.4 ± 4.9	8.1 ± 3.6	12.9 ± 1.8	60.74
PLA + 2% CNC	2894 ± 154	51.5 ± 2.3	2.5 ± 0.9	15.8 ± 10.7	60.55
PLA + 2% CNCd	2852 ± 45	47.5 ± 2.0	2.3 ± 0.3	12.8 ± 3	59.54
PCLdmf	302 ± 23	27.0 ± 3.4	505 ± 50	48.8 ± 5.6	−54.96
PCL + 2%CNC	358 ± 37	26 ± 1.3	135 ± 80	32.5 ± 1.9	−54.7
PCL + 2% CNCd	365 ± 94	27 ± 2.5	248 ± 114	30.1 ± 9.3	−54.26

## Data Availability

Data are contained within the article.

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
