# Peer review of "Effect of Polydopamine Coating of Cellulose Nanocrystals on Performance of PCL/PLA Bio-Nanocomposites"

_materials, 2023, doi:10.3390/ma16031087_

Round 1

Reviewer 1 Report

The present study focused on efficient effects of polydopamine- coating of cellulose nanocrystals 2 on performance of PCL/PLA bio-nanocomposites. First, the reagents were prepared using different 9 mixing protocols with solution- casting prepared masterbatches and wide range of blend ratios. Then, the products were characterized in details. The results are interesting. In my opinion, this manuscript can be published in this journal after major corrections:

-Many contents of abstracts should be removed. This section should focus on results and novelities.

-There are some typical error, mL instead of ml and….

- The resolution of TEM image is low

-What basis the Blend ratio of materials were used

- Standard division of Figs. 6 was missed.

-Recent advances about performance of novel nanocrystals should be added in text, for example: A) doi: https://doi.org/10.1016/j.apsusc.2022.155447, B: doi: https://doi.org/10.1016/j.memsci.2022.121099, C:doi: 10.1038/s41467-022-29962-6, D: doi: https://doi.org/10.1016/j.jmrt.2022.09.032 and E: doi: https://doi.org/10.1002/adfm.202202366

Author Response

Reviewer 1

Comments and Suggestions for Authors

The present study focused on efficient effects of polydopamine- coating of cellulose nanocrystals 2 on performance of PCL/PLA bio-nanocomposites. First, the reagents were prepared using different 9 mixing protocols with solution- casting prepared masterbatches and wide range of blend ratios. Then, the products were characterized in details. The results are interesting. In my opinion, this manuscript can be published in this journal after major corrections:

Point 1: -Many contents of abstracts should be removed. This section should focus on results and novelities.

Response: Abstract has been shortened and  improved accordingly

Point 2:-There are some typical error, mL instead of ml and….

Response: We have checked the text carefully and eliminated these errors

Point 3:- The resolution of TEM image is low

Response: We agree, but unfortunately, in spite of multiple samples analysis, we were not able to achieve better resolution. In spite of this fact, evaluation of tens of images indicates that with the PLA/CNCd masterbatch, a part of nanofiller is presented as „nanofiber-like“ arrays, whereas with the PCL/CNCd masterbatch such structures are absent (i.e. CNCd remains inside PCL inclusions). We believe that selected representative figures indicate this difference in spite of the mentioned poor resolution.

Point 4:-What basis the Blend ratio of materials were used

Response: We have used weight ratio, text was corrected accordingly.

Point 5:- Standard division of Figs. 6 was missed.

Response: According to our experience, the standard deviation of crystallinity of as-prepared PCL/PLA samples evaluated from DSC curves does not exceed 1.7 % and 2.9 % for PCL and PLA, respectively, and coefficient of variation is max. 2.9 % and 9.7 % for PCL and PLA, respectively (see J.Kratochvíl, I.Kelnar: Polymer Testing 47 (2015) 79-86). So the discussed differences in crystallinity, especially of PCL, exceed this scatter. Moreover, most important information is the low impact of nanoparticles used on crystallinities of both polymer components.

Point 6:-Recent advances about performance of novel nanocrystals should be added in text, for example: A) doi: https://doi.org/10.1016/j.apsusc.2022.155447, B: doi: https://doi.org/10.1016/j.memsci.2022.121099, C:doi: 10.1038/s41467-022-29962-6, D: doi: https://doi.org/10.1016/j.jmrt.2022.09.032 and E: doi: https://doi.org/10.1002/adfm.202202366

Response: In our opinion, introduction of the advances in the area of functional inorganic nanocrystals is out of the scope of this article dealing with „organic“ cellulose nanocrystals, moreover only applied as reinforcing nanofiller with favourable aspect ratio

Reviewer 2 Report

The work titled „Effect of polydopamine – coating od cellulose nanocrystals on performance of PCL/PLA bio-nanocomposites” requires some major correction befere resubmission to Materials Journal.

The Abstract seems to long, please check it. Some sentenced are too long, hard to read.

Line 18 – give full name of NF

Line 41-42 correct a style of the sentence

Influence of cellulosic fillers on polyester blend could be decribed in more details.

Line 58 – correct into „polymer blend ratio”

Experimental – The experiments should be described in the same order as in Results section.

Line 80 – what is MFR for PCL?

Line 87 – The Authors added dopamine, but then polydopamine was described. Is dopamine polymerized during modyfication? If yes, Authors should decribed this in more details.

Line 88 – remove „a” after „water”

Experimental part should be supported with graphical scheme of compostite blends preparation.

Line 105 – why crosshead speed 20mm/min was choosen? What standard for the test was applied?

Line 127 – 2.4. „Characterisation of blends structure”

Line 133 – add „(TEM)” as for the SEM

Line 181 – remove come after „may”

Fig. 1 The images should be presented with the same scale.

Table 1; There is „PLAdmf” and PCLdmf” but these abbrev. Are not used in the main text.

Fig. 5 correct „Sress” into „stress”

Line 307 – correct into „of all polymers ratio”

Line 309 – correct word „system”

Page 10. It is worh to mention that high E value is related to high content of PLA (generaly brittle material with high E)

Line 352 – „the systems”

Line 357 – do not start a sentence from abbreviations like e.g.

Page 14 – DMA

Due to the phase intelayers and some unhomogeneity the Tan Delta curves should be presented and disscused.

The DMA part is not cleary described. First of all, on the curves there are two peaks that need to be discussed. Authors can not take E” as Tg alone, add E’ and Tan Delta to the work.

The scale in Fig. 7 should be the same and start from 0°C.

Author Response

Reviever 2

The work titled „Effect of polydopamine – coating od cellulose nanocrystals on performance of PCL/PLA bio-nanocomposites” requires some major correction befere resubmission to Materials Journal.

Point 1:The Abstract seems to long, please check it. Some sentenced are too long, hard to read.

Response: Abstract has been shortened and improved accordingly

Point 2: Line 18 – give full name of NF

Response: Corrected in the new abstract version

Point 3: Line 41-42 correct a style of the sentence

Response: Sentence was re-written

Point 4: Influence of cellulosic fillers on polyester blend could be decribed in more details.

Response: Recent results in the area CNC- modified PLA/PCL have been highlighted

Point 5: Line 58 – correct into „polymer blend ratio”

Response: Corrected

Point 6: Experimental – The experiments should be described in the same order as in Results section.

Response: The order was changed accordingly

Point 7: Line 80 – what is MFR for PCL?

Response: MFR has been added.

Point 8: Line 87 – The Authors added dopamine, but then polydopamine was described. Is dopamine polymerized during modyfication? If yes, Authors should decribed this in more details.

Response: corrected accordingly

Point 9: Line 88 – remove „a” after „water”

Response: Corrected to “and”

Point 10: Experimental part should be supported with graphical scheme of compostite blends preparation.

Response: In our opinion, the conventional preparation method, applied by many authors need not be supported by graphical sheme

Point 11: Line 105 – why crosshead speed 20mm/min was choosen? What standard for the test was applied?

Response: Thank for this remark, the speed was incorrect, the standart applied was added.

Point 12: Line 127 – 2.4. „Characterisation of blends structure”

Response: Corrected accordingly

Point 13: Line 133 – add „(TEM)” as for the SEM

Response: Corrected accordingly

Point 14: Line 181 – remove come after „may”

Response: Corrected accordingly

Point 15: Fig. 1 The images should be presented with the same scale.

Response: In our opinion, the image a) with lower magnification better describe the dimensions of lamellar arrays formed in PLA nanocomposite

Point 16: Table 1; There is „PLAdmf” and PCLdmf” but these abbrev. Are not used in the main text.

Response: Explanation has been added to Table 1 caption

Point 17: Fig. 5 correct „Sress” into „stress”

Response: Corrected accordingly

Point 18: Line 307 – correct into „of all polymers ratio”

Response: Corrected accordingly

Point 19: Line 309 – correct word „system”

Response: Corrected accordingly

Point 20: Page 10. It is worh to mention that high E value is related to high content of PLA (generaly brittle material with high E)

Response: Corrected accordingly

Point 21 :Line 352 – „the systems”

Response: Corrected accordingly

Point 22: Line 357 – do not start a sentence from abbreviations like e.g.

Response: Corrected accordingly

Point 23: Page 14 – DMA

Due to the phase intelayers and some unhomogeneity the Tan Delta curves should be presented and disscused.

The DMA part is not cleary described. First of all, on the curves there are two peaks that need to be discussed. Authors can not take E” as Tg alone, add E’ and Tan Delta to the work.

Response: In our opinion, the main purpose of DMA in this work was to demonstrate the effect of CNC and its modification on Tg of polymer components. Therefore, to our experience, presentation of thermal dependence of E” is sufficient. The modulus of all systems studied was more precisely characterized by tensile testing.

Point 24: The scale in Fig. 7 should be the same and start from 0°C.

Response: Corrected accordingly

Reviewer 3 Report

Manuscript ID materials-2156701

The subject addressed in the present manuscript is of potential interest to Materials readers. However, the manuscript presents serious flaws.

The abstract needs deep improvement. First, as it was presented do not fit the author's guidelines, being too long. It is confusing and does not have the necessary information with enough quality to understand the focus of the work and the main achievements. Must be totally rewritten.

The manuscript text did not care. Too many errors and typos are found in the manuscript, and this should not happen. Thus, the entire text should be revised.

Also, the way the manuscript was built is confusing. Many examples can be found. For example, something is missing in the first sentence of the introduction “It is now well accepted that the impact of nanofillers (NF) on the performance of polymers is greater mainly in the case of a multicomponent polymer matrix [1-4].” Nanofillers have an impact on the performance of polymers or composites made with polymers? It is not easy to follow the ideas and the work, as well as the discussion of the results. For example, authors start by discussing first a figure presented in supplementary material, followed by figure 2. Only after some paragraphs figure 1 is discussed. This does not make sense.

The details given in the materials and methods section are not enough or are very confusing. For example in lines 95-96: “Two sets of samples differing in addition of CNC via PCL- or PLA-masterbatch with 80/20,60/40,40/60and 80/20 PCL/PLA ratios were prepared”. Is not possible to understand the ratios used. Never were tested only PCL and PLA, neat? If not should be included as a reference.

The quality of the TEM images mainly figures 1b and 1c is very poor. Must be improved in order to take some information from these images.

All the SEM discussion is not correct in my opinion. Authors seem to discuss the “dark zones” as particles inside the matrix. In my opinion, the dark zones are air bubbles. Thus, all the discussion in section 3.2 is not correct and needs to be redone.

The particle size presented in table 1 was obtained from which technique? If was from SEM images, in my opinion, and by the previously exposed, the analysis is wrong.

In the end, mainly in section 3.5 and in the conclusions, I did not understand which is the mixture state of the composites. In line 452 authors stated “Certain decrease of Tg of PLA in all systems is a consequence of PCL dissolved in the PLA-rich phase” and in conclusions, line 473, “In the systems with dominating bicontinuous structure (60/40 and 40/60 ratio)”. It is necessary to understand the state of the dispersion/mixture of the composites.

Author Response

Reviewer 3

The subject addressed in the present manuscript is of potential interest to Materials readers. However, the manuscript presents serious flaws.

Point 1: The abstract needs deep improvement. First, as it was presented do not fit the author's guidelines, being too long. It is confusing and does not have the necessary information with enough quality to understand the focus of the work and the main achievements. Must be totally rewritten.

Response: Abstract has been improved accordingly

Point 2: The manuscript text did not care. Too many errors and typos are found in the manuscript, and this should not happen. Thus, the entire text should be revised.

Response: Thorough revision of the text has been done

Point 3: Also, the way the manuscript was built is confusing. Many examples can be found. For example, something is missing in the first sentence of the introduction “It is now well accepted that the impact of nanofillers (NF) on the performance of polymers  is greater mainly in the case of a multicomponent polymer matrix [1-4].” Nanofillers have an impact on the performance of polymers or composites made with polymers? It is not easy to follow the ideas and the work, as well as the discussion of the results. For example, authors start by discussing first a figure presented in supplementary material, followed by figure 2. Only after some paragraphs figure 1 is discussed. This does not make sense.

Response: We believe that we have improved most of incorrect formulations; the first mentioning of Fig. 2 is a mistake, it was corrected to Fig. 1 a

Point 4: The details given in the materials and methods section are not enough or are very confusing. For example in lines 95-96: “Two sets of samples differing in addition of CNC via PCL- or PLA-masterbatch with 80/20,60/40,40/60and 80/20 PCL/PLA ratios were prepared”. Is not possible to understand the ratios used. Never were tested only PCL and PLA, neat? If not should be included as a reference.

Response: The description was improved, PLA/PCL ratios were corrected to weight ratios. The parameters of neat components and related nanocomposites are presented in Table S2

Point 5: The quality of the TEM images mainly figures 1b and 1c is very poor. Must be improved in order to take some information from these images.

Response: We agree, but unfortunately, in spite of multiple samples analysis, we were not able to achieve better resolution. In spite of this fact, evaluation of tens of images indicates that with the PLA/CNCd masterbatch, a part of nanofiller is presented as „nanofiber-like“ arrays, whereas with PCL/CNCd masterbatch, such structures are absent (i.e. CNCd remains inside PCL inclusions). We believe that selected representative figures  indicate this difference in spite of the mentioned poor resolution.

Point 6: All the SEM discussion is not correct in my opinion. Authors seem to discuss the “dark zones” as particles inside the matrix. In my opinion, the dark zones are air bubbles. Thus, all the discussion in section 3.2 is not correct and needs to be redone.

Response: All unetched samples were voids-free. Therefore, we consider the “dark zones” to be PLA subinclusions inside PCL threads just visualized by hydroxide (20% NaOH)-etching (in addition to etching of PLA threads)

Point 7: The particle size presented in table 1 was obtained from which technique? If was from SEM images, in my opinion, and by the previously exposed, the analysis is wrong.

Response: We have used etching techniques applied successfully in many similar studies; the particle size evaluation using semi-automatic device is mentioned in section 2.4.

Point 8: In the end, mainly in section 3.5 and in the conclusions, I did not understand which is the mixture state of the composites. In line 452 authors stated “Certain decrease of Tg of PLA in all systems is a consequence of PCL dissolved in the PLA-rich phase” and in conclusions, line 473, “In the systems with dominating bicontinuous structure (60/40 and 40/60 ratio)”. It is necessary to understand the state of the dispersion/mixture of the composites.

Response: We believe that the mixture state (matrix/inclusions and bicontinuos structures) is obvious from SEM images of respective systems. For clarity, references to corresponding images have been added.

Round 2

Reviewer 1 Report

Authors revised manuscript carefully. The final version  of this manuscript can be published in this journal.

Reviewer 2 Report

In my opinion the work have not been sufficiently corrected.

"Point 16: Table 1; There is „PLAdmf” and PCLdmf” but these abbrev. Are not used in the main text." 

- The acronym can be used instead of longer description that Authors used e.g. in line 201.

"The DMA part is not cleary described. First of all, on the curves there are two peaks that need to be discussed. Authors can not take E” as Tg alone, add E’ and Tan Delta to the work."

- These two peaks in loss modulus curves still have not been described properly. You have a polymer blend so which Tg is for PCL and which is for PLA? 
Moreover loss modulus is a method to measure dissipation energy, so Authors should explain how it is related to Tg? 

Reviewer 3 Report

Manuscript ID materials-2156701

After the authors revise their manuscript, many of my concerns are still present. Many typos and errors continue present in the manuscript text. The structure of the manuscript is the same, thus confusing.

Some questions perhaps were not understood by the authors. For example, do nanofillers have an impact on the performance of polymers or composites made with polymers? It is not easy to follow the ideas and the work, as well as the discussion of the results.

The details given in the materials and methods section are not enough or are very confusing. For example in lines 95-96: “Two sets of samples differing in addition of CNC via PCL- or PLA-masterbatch with 80/20,60/40,40/60and 80/20 PCL/PLA ratios were prepared”. Is not possible to understand well the ratios used. This question was raised previously and the authors did not change it. 80/20 is repeated for example or not poorly explained.

If it is not possible to obtain TEM images with better resolution, the images should be removed from the manuscript. As the images are presented do not give any useful information.

The explanation given by the authors about SEM discussion should be supported. Thus, images without etching should be provided and the effect of the etching (which polymer was removed and the removal extension) introduced in the discussion.

The particle size presented in table 1 was obtained from which technique? This needs clarification.

For me is not obvious the mixture state of the composites is as stated by the authors: “is obvious from SEM images of respective systems”.
